# Constructing asymmetric double-atomic sites for synergistic catalysis of electrochemical $CO_2$ reduction

Jiqing Jiao [1,6] ✉, Qing Yuan[2,6], Meijie Tan[1], Xiaoqian Han[1], Mingbin Gao [3], Chao Zhang[1], Xuan Yang [2] ✉, Zhaolin Shi[1], Yanbin Ma[1], Hai Xiao [4], Jiangwei Zhang [5] ✉ & Tongbu Lu [1] ✉

Elucidating the synergistic catalytic mechanism between multiple active centers is of great significance for heterogeneous catalysis; however, finding the corresponding experimental evidence remains challenging owing to the complexity of catalyst structures and interface environment. Here we construct an asymmetric $TeN_2$−$CuN_3$ double-atomic site catalyst, which is analyzed via full-range synchrotron pair distribution function. In electrochemical $CO_2$ reduction, the catalyst features a synergistic mechanism with the double-atomic site activating two key molecules: operando spectroscopy confirms that the Te center activates $CO_2$, and the Cu center helps to dissociate $H_2O$. The experimental and theoretical results reveal that the $TeN_2$−$CuN_3$ could cooperatively lower the energy barriers for the rate-determining step, promoting proton transfer kinetics. Therefore, the $TeN_2$−$CuN_3$ displays a broad potential range with high CO selectivity, improved kinetics and good stability. This work presents synthesis and characterization strategies for double-atomic site catalysts, and experimentally unveils the underpinning mechanism of synergistic catalysis.

Heterogeneous catalysis is the preferred mode to design new and environment-friendly methodologies for green chemistry, among which electrochemical $CO_2$ reduction (CO2RR) is an important approach to carbon neutrality[1–4]. The CO2RR process takes place at the catalytic interface with complicated proton/electron transfer[5–7], and elucidating the synergistic mechanism between the active sites is crucial for high-efficiency catalysis[8,9]. Among the different pathways involved in CO2RR, $CO_2$-to-CO conversion is the central one[10,11]. Single-atomic site catalysts (SACs) with well-defined active sites and tunable coordination environments represent an ideal model for understanding the structure−performance relationship[12,13]. In the past few years, a number of M−N−C catalysts (M=Ni[14–16], Fe[17,18], Co[19], Zn[20,21], Mn[22], etc.) have been developed for CO2RR. However, the structurally simple SACs may not be ideal for activating the CO2RR process (a typical multi-molecule-participating reaction), and thus it is rather difficult to break the linear scaling relationships between the adsorption energies of reaction intermediates[23].

Compared with SACs, double-atomic site catalysts (DACs) are structurally more complicated, with double active centers at play in close vicinity; for the CO2RR process (with multiple molecules

[1]MOE International Joint Laboratory of Materials Microstructure, Institute for New Energy Materials and Low Carbon Technologies, School of Materials Science and Engineering, Tianjin University of Technology, Tianjin 300384, China. [2]Hubei Key Laboratory of Material Chemistry and Service Failure, School of Chemistry and Chemical Engineering, Huazhong University of Science and Technology, 1037 Luoyu Road, Wuhan 430074, China. [3]National Engineering Research Center of Lower-Carbon Catalysis Technology, Dalian National Laboratory for Clean Energy, Dalian Institute of Chemical Physics, Chinese Academy of Sciences, Dalian 116023, China. [4]Department of Chemistry, Tsinghua University, Beijing 100084, China. [5]Science Center of Energy Material and Chemistry, College of Chemistry and Chemical Engineering, Inner Mongolia University, Hohhot 010021, P. R. China. [6]These authors contributed equally: Jiqing Jiao, Qing Yuan. ✉e-mail: jiaojiqing101@163.com; xuanyang@hust.edu.cn; zjw11@tsinghua.org.cn; lutongbu@tjut.edu.cn

participating and multiple electron transfers involved), the DACs are expected to feature synergistic catalysis while inheriting the advantage of high atom utilization typically for SACs. Recently, a range of metal-based DACs have been developed for $CO_2RR$ with higher performances than corresponding SACs, such as Fe−N sites with cobalt phthalocyanine[24], Ni−Zn bimetal sites[25], neighboring Zn/Co monomers[26], isolated diatomic Ni−Fe sites[27], N-bridged Co−N−Ni[28], NiSn atomic pair[23], Ni/Cu dual sites[29], Fe/Ni−N[18]. These DACs usually employ transition metals as the active centers anchored by N atoms. In addition, the aforementioned DACs usually have symmetric configurations, which would lead to a high free energy for water dissociation, and thus a sluggish proton transfer kinetics[21]. Inspired by the stereo-specific catalysis in biomolecules (such as enzymes), in a previous work we designed and prepared a DAC featuring asymmetric $Cu^0$−$Cu^+$ pairs; the synergy within the Cu atom pair allows for activation of both $CO_2$ and $H_2O$, collectively resulting in a high performance[30].

Here, we select a semimetal (tellurium, Te)[31] and a transition metal (copper, Cu) to construct the DAC. The non-planar, asymmetric structure of $TeN_2$−$CuN_3$ site was characterized by combining the full-range synchrotron pair distribution function (PDF) and synchrotron radiation-based X-ray absorption spectroscopy. For $CO_2RR$, the DAC displays a broad potential range of high $FE_{CO}$ (>90%), improved reaction kinetics, and good stability. Theoretical calculations revealed that the $TeN_2$−$CuN_3$ sites could synergistically lower the energy barriers for the rate-determining step, thus effectively promoting the proton transfer kinetics. Operando attenuated total reflection surface-enhanced infrared absorption spectroscopy (ATR-SEIRAS) revealed that in the asymmetric $TeN_2$−$CuN_3$, the Te center activates $CO_2$, and the Cu center helps to dissociate $H_2O$, thus promoting the $CO_2RR$ process via a synergistic mechanism.

## Results

### Synthesis and characterizations of $TeN_2$−$CuN_3$ DAC

For the synthesis of $TeN_2$−$CuN_3$ DACs, a sacrificial template method was employed in combination with a double-solvent impregnation method. Uniform Te nanowires (NWs) were synthesized first (following the method in previous reports including our work[30]; the details can be found in Supplementary Information). The Te NWs were then coated with ZIF-8 frameworks, forming a core−sheath composite. Cu ions were then introduced into the pores of ZIF-8 via the double-solvent method. The core−sheath composite was redispersed in n-hexane, and a prescribed amount of $CuCl_2$ solution (in methanol) was added under room temperature; after stirring for 12 h, the solid was separated via centrifugation; then the solid as the precursor was heated to 1000 °C for 3 h under $N_2$, yielding the final product (Fig. 1a). During the pyrolysis process at 1000 °C, the Cu atoms were anchored on the resulting N-doped carbon support. The Te atoms were evaporated from the inside Te nanowires to the outside porous N-doped carbon support. As Cu sites are mainly anchored by the N atoms in N-doped carbon support, the subsequent Te−N coordination interaction can let the Te atoms anchored near Cu sites to generate $TeN_2$−$CuN_3$ DACs.

The characterization results on the morphology and composition of the as-synthesized catalyst are summarized in Fig. 1b−g. Figure 1b shows the transmission electron microscopy (TEM) image of the uniform Te NWs with a diameter of 5 nm. Figure 1c, d shows the typical TEM images of the core−sheath structured Te@ZIF-8 and Cu-containing Te@ZIF-8. Figure 1e shows the porous structure of the product obtained after pyrolysis, with no nanoparticles and clusters observed on the carbon support. Supplementary Fig. 1−3 summarizes the X-ray diffraction (XRD)[32,33] patterns for the different products in each step; after pyrolysis, the characteristic peaks for Te NWs (JCPDS PDF 65−3370) disappeared, and no signals of Cu were detected. In the X-ray photoelectron spectroscopy (XPS) data (Supplementary Fig. 4), no noticeable signals of Te and Cu were found, owing to their low

loadings. In addition, comparison samples with various Cu loadings were also prepared by controlling the amount of $CuCl_2$ introduced, and the actual loadings of Te and Cu were determined via inductively coupled plasma mass spectroscopy (ICP-MS), as listed in Supplementary Table 1 and Table 2.

Furthermore, the aberration-corrected high-angle annular dark-field scanning transmission electron microscopy (HAADF-STEM) images (Fig. 1f and Supplementary Fig. 5) show evenly distributed bright spots with a high density, indicating that Te and Cu are dispersed atomically on the N-doped porous carbon (NC) support; the red circles highlight the bright spots that are in close proximity, suggesting the formation of diatomic sites. No nanoparticles or clusters were found, which is in good agreement with the XRD patterns and XPS results. Figure 1g shows three typical spot pairs. We also conducted statistical analysis on the separation distance in the spot pairs by counting > 100 pairs. The spot pairs can be clearly identified by examining the intensity profiles (Fig. 1g, bottom), and 78% of the spot pairs were found as diatomic sites (with a separation distance smaller than 0.33 nm). The HAADF-STEM images of $TeN_3$ and $CuN_4$ can be found in Supplementary Fig. 6 and 7, respectively. The elemental distribution is revealed via energy-dispersive X-ray spectroscopy (EDS) mapping; Supplementary Fig. 8 shows that Te and Cu elements were evenly dispersed on the N-doped carbon support. The $N_2$ adsorption−desorption isotherm (Supplementary Fig. 9) reveals that the DAC has a larger Brunauer−Emmett−Teller (BET) surface area. As shown in the Raman spectra (Supplementary Fig. 10), the D peak (at 1345 $cm^{-1}$) is due to out-of-plane vibrations attributed to the presence of structural defects, whereas the G peak (at 1580 $cm^{-1}$) comes from the in-plane vibrations of sp2 bonded carbon atoms[23,34]. Therefore, the increased $I_D/I_G$ ratio indicates that the formation of $TeN_2$−$CuN_3$ DAC induces more structural defects.

### XAFS and PDF analysis of $TeN_2$−$CuN_3$ DAC

The Te and Cu atoms are distinctly different in size and electronegativity, and therefore on the N-doped C support these two atoms are expected to have different coordination configurations and electronic structures. In this regard, we conducted synchrotron radiation-based X-ray absorption fine structure (XAFS) at the Te K-edge and the Cu K-edge for different samples (DAC, and SACs of Te and Cu), and the corresponding references including Te foil, $TeO_2$, Cu foil, $Cu_2O$, CuO and copper(II) phthalocyanine (CuPc). The X-ray absorption near-edge structure (XANES) profiles for $TeN_2$−$CuN_3$ DAC and $TeN_3$ SAC are similar, and the edge positions are between those for Te and $TeO_2$, because N has a higher electronegativity than Te. In addition, their white-line intensities are significantly higher than that of Te foil (Fig. 2a). Notably, compared with the $TeN_2$−$CuN_3$ DAC, the $TeN_3$ SACs show an edge position slightly positively shifted towards that of $TeO_2$, indicating a higher oxidation state for the Te atoms in $TeN_3$. Similarly, the white-line intensity for $CuN_4$ is distinctly higher than that of $TeN_2$−$CuN_3$ (Fig. 2d), which reveals that the Cu atoms in $CuN_4$ have a higher oxidation state. To determine the exact oxidation states of Te and Cu in the DACs and SAs, the XANES data were fitted using the linear combination fitting (LCF) method (Supplementary Fig. 11, Supplementary Table 3 and Table 4). The valence state of Te and that of Cu for $TeN_2$−$CuN_3$ DACs are lower than that for $TeN_3$ and $CuN_4$ SAs, indicating a higher electron density at the catalytic site. These XANES data confirm that the Te and Cu atoms in the DAC have different oxidation states than in the corresponding SACs, hinting at the altered coordination environments and electronic structures of Te and Cu in the DAC.

The detailed parameters of local atomic structure including coordination numbers (CNs) and bond lengths were investigated via EXAFS. The fitting quantitative χ(R) space and Fourier-transform (FT) k2-weighted function χ(k) spectra were also performed to investigate local atomic structure and to further derive the CNs of Te in

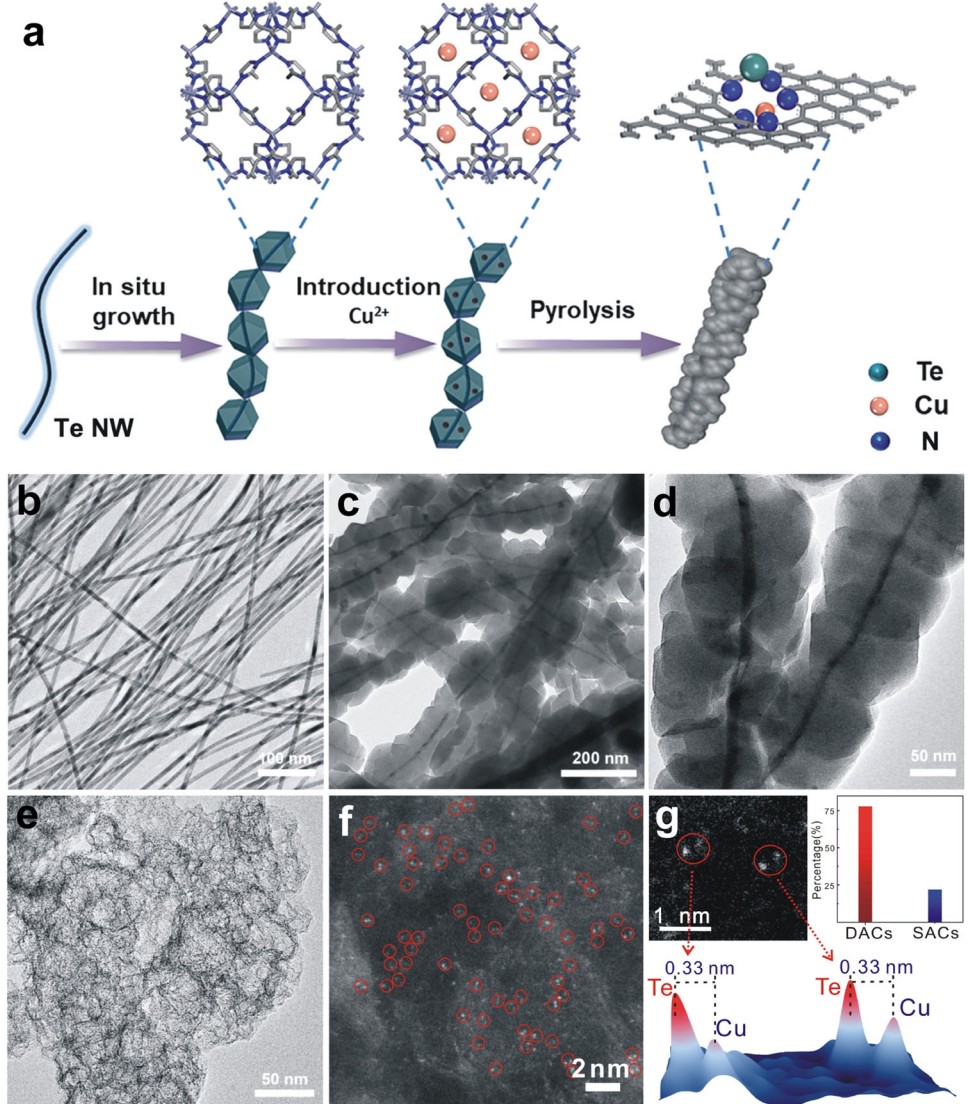

**Fig. 1 | Synthesis and characterizations of TeN$_2$–CuN$_3$ DAC. a** Schematic illustration for the synthesis of TeN$_2$–CuN$_3$ DAC. **b–e** TEM images for Te nanowires (**b**), core–sheath structure of Te@ZIF-8 (**c**), Te@ZIF-8 with Cu ions (**d**), and TeN$_2$–CuN$_3$ DAC (**e**). **f** HAADF-STEM image of TeN$_2$–CuN$_3$ DAC. **g** Typical spot pairs for TeN$_2$–CuN$_3$ DAC (top left), statistical analysis for double sites and single sites (top right), corresponding 3D intensity profiles of two pairs of DAC sites (bottom).

TeN$_2$–CuN$_3$. The data in Supplementary Fig. 12 are consistent with the fitting results in Supplementary Table 5; the CN of Te–N bond is close to 2.0 for TeN$_2$–CuN$_3$, whereas for TeN$_3$ SACs, the CN of Te–N bond is 3.0 (Supplementary Fig. 13 and Supplementary Table 6). As shown in Fig. 2b, the radial distance space spectra χ(R) for the TeN$_3$ and TeN$_2$–CuN$_3$ show a major peak at ~1.85 Å and 1.91 Å, respectively, which is attributed to the Te–N scattering path. The Te–N–C scattering paths for TeN$_3$ and TeN$_2$–CuN$_3$ are located at ~2.75 Å and 2.85 Å, respectively. Clearly, the scattering paths of Te–N and Te–N–C for TeN$_2$–CuN$_3$ are larger than those for TeN$_3$. In addition, Fig. 2b shows that the Te–Te path (2.92 Å) for Te foil is close to the Te–N–C path for TeN$_2$–CuN$_3$, and we conducted data fitting to further distinguish the two paths. As shown in Fig. 2c, the fitting results for Te–N and Te–N–C paths agree well with the experimental data, and the χ(R) intensity for Te–N–C is far lower than that for Te–Te, which collectively indicates that Te is atomically dispersed on the support. Likewise, in the Cu K-edge χ(R) space spectra (Fig. 2e), the Cu–N and Cu–N–C paths for TeN$_2$–CuN$_3$ are slightly shifted to ~1.87 Å and 2.60 Å, respectively. The CN was determined to be 3.0. The quantitative χ(R) and χ(k) space spectra fitting were also performed (Supplementary Fig. 14 and Supplementary Table 7). The Cu–N and Cu–N–C scattering paths for CuN$_4$ (the CN of

Cu–N bond is close to 4.0) are located at ~1.77 Å and 2.55 Å, respectively (Supplementary Fig. 15 and Supplementary Table 8). Compared with the cases in SACs, both Te and Cu atoms in the double-atomic sites have lower CNs with N atoms, which induces the elongation of the Te–N, Cu–N, Te–N–C, and Cu–N–C paths.

In order to further unveil the detailed atomic structures of Te and Cu in the diatomic sites, we performed a pair distribution function G(r) (PDF (G(r)) analysis. The PDF Rietveld refinement result for partial Cu–Te paths of TeN$_2$–CuN$_3$ is shown in Fig. 2f. The peak at ~3.29 Å is attributed to the Te–Cu path; the other peaks at longer distances are due to the multiple scattering paths (>5 Å). The peaks in Supplementary Fig. 16 labeled from A–E can be ascribed to different atomic pair distances for TeN$_2$–CuN$_3$ catalyst[35,36]. The peaks labeled A correspond to C–N and C–C in N-doped C support; the peak at ~1.72 Å (labeled B), to Cu–N; the peak at ~2.15 Å (labeled C), to Te–N; the peak at ~1.88 Å, to Cu–N and Te–N (resulting from the asymmetric structure); the peaks at ~3.16 Å and 3.52 Å (labeled D), to Cu/Te–N–C; the peak at ~3.29 Å (labeled E), to Te–Cu (Supplementary Table 9). Furthermore, wavelet transform (WT) of χ(k) is an intuitive way to demonstrate the bonding features of TeN$_2$–CuN$_3$ in comparison to the TeN$_3$, Te foil and TeO$_2$ references (Fig. 2g). Two signals of Te–N and Te–N–C located

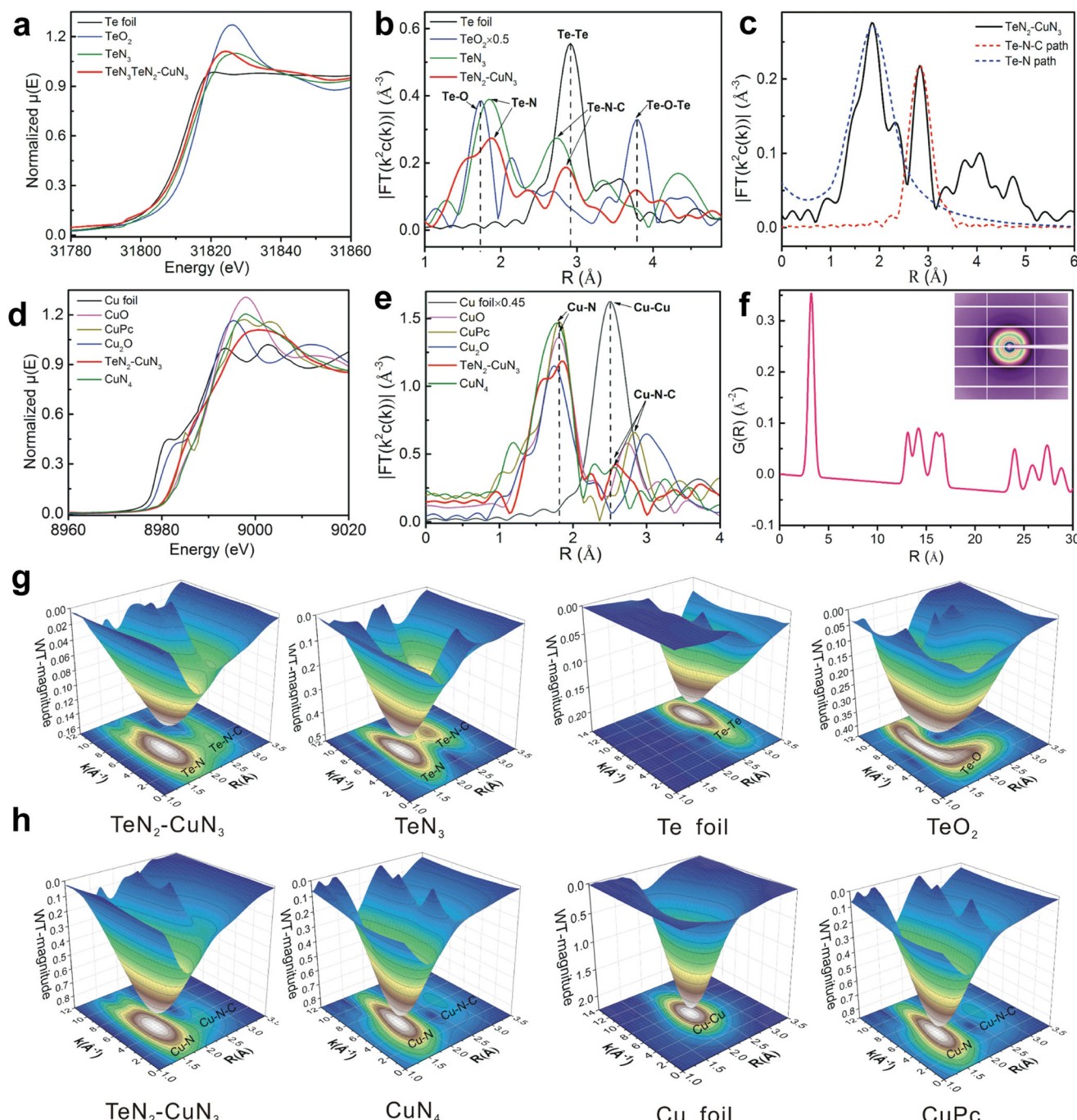

**Fig. 2 | XAFS and PDF analysis for TeN$_2$−CuN$_3$ DAC. a**, **d** Normalized XANES $\chi(E)$ spectra of Te and Cu. **b**, **e** Radial distance $\chi(R)$ space spectra of Te and Cu. **c** Fitting results for Te−N and Te−N−C paths of TeN$_2$−CuN$_3$ DAC. **f** PDF Rietveld refinement result for partial Cu−Te paths of TeN$_2$−CuN$_3$ DAC; (inset) 2D scattering image of DAC. **g**, **h** WT of $\chi(k)$ of Te and Cu for TeN$_2$−CuN$_3$ DAC and other samples.

respectively at [$\chi(k)$, $\chi(R)$] of [4.7, 1.91] and [4.0, 2.85] were found. The $\chi(R)$ values (increased compared with the signals of TeN$_3$ ([6.2, 1.85] and [7.4, 2.75])) are due to the formation TeN$_2$−CuN$_3$ DAC. For the Cu K-edge WT of $\chi(k)$ spectra for TeN$_2$−CuN$_3$, two signals of Cu−N and Cu−N−C located respectively at [5.0, 1.87] and [3.3, 2.60] were found. The $\chi(R)$ values are also larger than those for Cu−N and Cu−N−C ([6.0, 1.77] and [5.9, 2.55]) in CuN$_4$. Clearly, compared with the case in SACs, the coordination structures of Te and Cu are significantly different in the DAC. To sum up, using both XAFS and PDF ($G(r)$ Rietveld analyses), we can now depict the structural characteristics of the DAC. In the TeN$_2$−CuN$_3$ DAC, the distance between Te and Cu atoms is approximately 3.3 Å with no bond formed in between, which is consistent with

HAADF-STEM image (Fig. 1f) and XAFS analysis (Fig. 2); specifically, the Te and Cu atoms are closely immobilized on the support, coordinated with two and three N atoms, respectively. To sum up, the CNs for Te and Cu are both lower than those in the corresponding SACs, and the resulting TeN$_2$−CuN$_3$ site features a distorted, asymmetric structure.

## Assessment of CO$_2$RR performances

The CO$_2$RR performances of TeN$_2$−CuN$_3$, TeN$_3$, CuN$_4$, and NC were assessed using a standard three-electrode setup in an H-type cell, with CO$_2$-saturated 0.1 M KHCO$_3$ solution as the electrolyte. All potentials in this work were reported with respect to the reversible hydrogen electrode (RHE). All the products were monitored via online gas

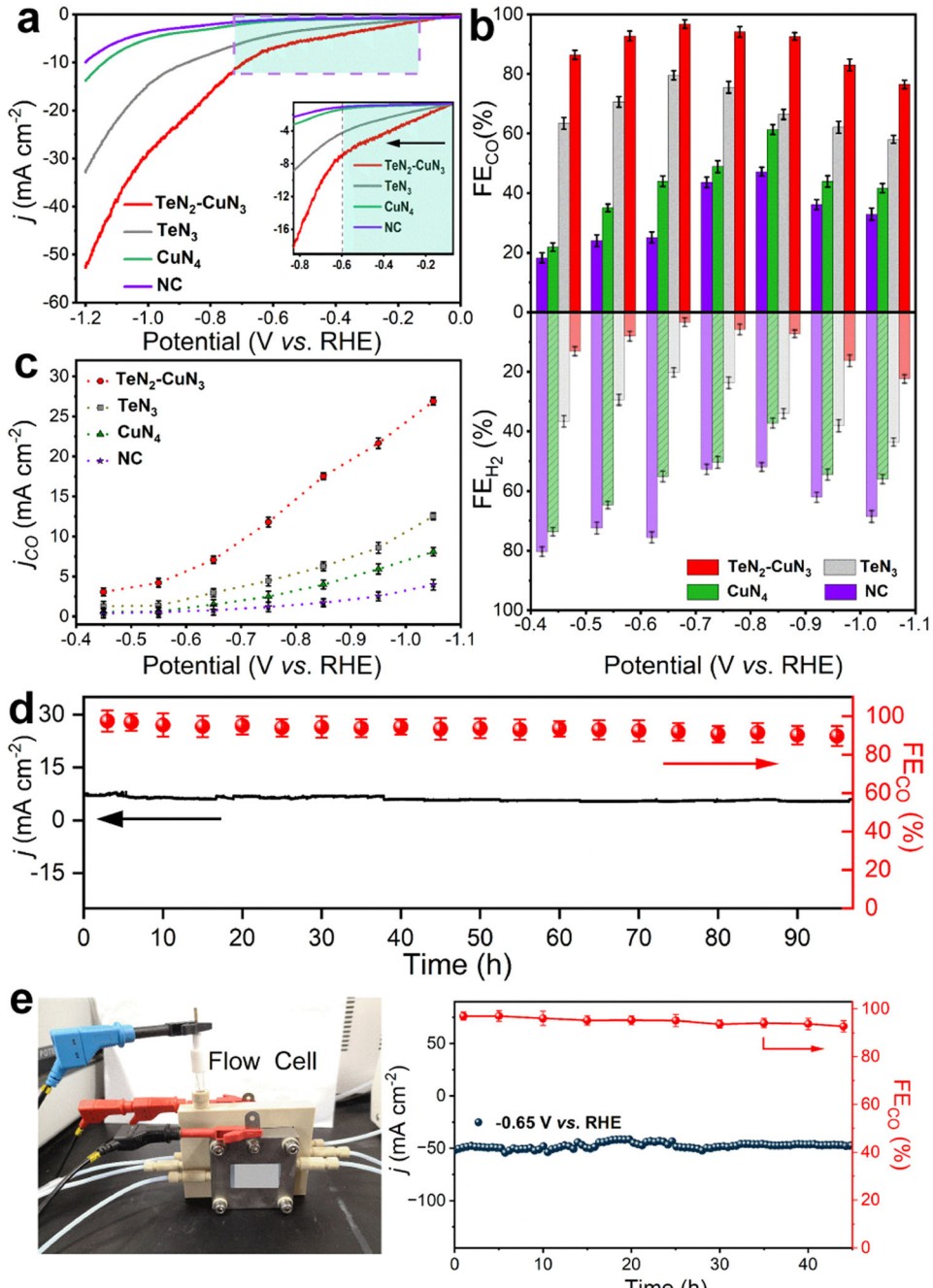

**Fig. 3 | The CO₂RR performances of TeN₂–CuN₃, TeN₃, CuN₄, and NC. a** LSV curves with *iR* correction (*i*, current; *R*, resistance: 9.0 ± 0.6 ohm; electrode surface area: 0.25 cm²). **b** Faradaic efficiencies (upper panel: FE$_{CO}$; bottom panel: FE$_{H2}$) and (**c**) *j*$_{CO}$ of different samples for applied potentials. Error bars stand for standard deviations. **d** Long-term stability and corresponding FE$_{CO}$ of TeN₂–CuN₃ DAC. **e** The photograph of the flow cell (left) and long-term test of TeN₂–CuN₃ at −0.65 V (vs. RHE) (right). Error bars represent the standard deviation of three independent measurements.

chromatography (GC) (Supplementary Fig. 17) and ¹H nuclear magnetic resonance (NMR) spectroscopy (Supplementary Fig. 18). As shown in the linear sweep voltammetry (LSV) curves (Fig. 3a and Supplementary Fig. 19), TeN₂–CuN₃ displays the highest total current density in the tested potential range. Compared with those for the other reference samples, the onset potential for TeN₂–CuN₃ is significantly less negative (inset of Fig. 3a), indicating a higher CO₂RR catalytic activity.

As shown in Fig. 3b, the FE$_{CO}$ (upper panel) and FE$_{H2}$ (bottom panel) for the samples TeN₂–CuN₃, TeN₃, CuN₄, and NC were evaluated for different cathodic potentials. It is noticeable that the TeN₂–CuN₃

gives a high selectivity (FE$_{CO}$ > 90%) over a broad potential range from −0.55 V to −0.85 V (vs. RHE), with a maximum FE$_{CO}$ of 98% at −0.65 V (*vs.* RHE). By contrast, the FE$_{CO}$ for TeN₃ reaches the maximum of 80% at −0.65 V (vs. RHE), and the FE$_{CO}$ for CuN₄ reaches the maximum of 60% at a more negative potential (−0.85 V (vs. RHE)); in combination with the LSV data, it can be seen that TeN₃ outperforms CuN₄ in CO₂-to-CO conversion. The NC support displays the lowest FE$_{CO}$ and current density among all the samples, indicating the catalytic activities of TeN₂–CuN₃, TeN₃, and CuN₄ come from the Te and Cu atoms. Figure 3c summarizes the data on partial current density for CO (*j*$_{CO}$); TeN₂–CuN₃ delivers the highest *j*$_{CO}$, reaching 8.0 mA cm⁻² at −0.65 V

(vs. RHE), which is about 4 and 8 times higher than that for $TeN_3$ and $CuN_4$, respectively. In particular, the $j_{CO}$ for $TeN_2$–$CuN_3$ is larger than the sum of the $j_{CO}$ values for $TeN_3$ and $CuN_4$ over the entire tested potential range, suggesting a synergistic effect between Te and Cu centers in the DAC.

A series of comparison samples with different Cu loadings were synthesized, and their $CO_2RR$ performances were investigated (Supplementary Fig. 20). The highest performance (in terms of $FE_{CO}$ and $j_{CO}$) was found for a Cu loading of 0.53 wt%; when the Cu loading is lower, the catalytic performance becomes inferior, owing probably to the smaller amount of active centers; when the Cu loading is higher, HER becomes more pronounced. Furthermore, as shown in (Supplementary Fig. 21), the Tafel slope for the $TeN_2$–$CuN_3$ was determined to be 65.2 mV dec$^{-1}$, which is much smaller than those for $TeN_3$ (88.1 mV dec$^{-1}$) and $CuN_4$ (190.6 mV dec$^{-1}$), revealing a favorable reaction kinetics for CO generation on $TeN_2$–$CuN_3$ DAC.

In addition, electrochemical impedance spectroscopy (EIS) was conducted (Supplementary Fig. 22). The smallest semicircle diameter for $TeN_2$–$CuN_3$ indicates the fastest surface charge transfer, which also hints at a favorable reaction kinetics. The electrochemically active surface area (ECSA) was determined using the double-layer capacitance ($C_{dl}$) method, and the results show that the encapsulated structure has a larger ECSA (Supplementary Fig. 23) The ECSA of $TeN_2$–$CuN_3$, $TeN_3$, and $CuN_4$ catalysts are determined as 700.4 cm$^2$, 676.2 cm$^2$ and 552.1 cm$^2$, respectively. The ECSA of $TeN_2$–$CuN_3$ is slightly larger than those for the other SAC samples. Thus, the improved $CO_2RR$ activity of $TeN_2$–$CuN_3$ is attributed primarily to synergistic effect between Te and Cu active sites. As shown in Supplementary Fig. 24, $TeN_2$–$CuN_3$ DAC displayed a TOF around 24080 h$^{-1}$ at −0.65 V (vs. RHE), which is 2.5 and 9.3 times higher than that of $TeN_3$ (9720 h$^{-1}$) and $CuN_4$ (2590 h$^{-1}$), respectively. In the applied potentials, the calculated TOFs for $TeN_2$–$CuN_3$ are even higher than the sum of the TOFs of $TeN_3$ and $CuN_4$, which revealed that the $TeN_2$–$CuN_3$ DAC displayed an intrinsically higher $CO_2RR$ activity than $TeN_3$ and $CuN_4$. The above analysis shows that the kinetics for $TeN_3$ and $CuN_4$ has been improved for $TeN_2$–$CuN_3$ via the synergistic catalytic mechanism. Moreover, the $TeN_2$–$CuN_3$ DAC also displayed good stability; the $j_{CO}$ and $FE_{CO}$ (>90%) were nearly unchanged after electrolysis for 96 h at −0.65 V (vs. RHE). (Fig. 3d) Furthermore, the $CO_2RR$ performance of $TeN_2$–$CuN_3$ DAC was measured using flow-cell configuration. As shown in Fig. 3e, the stability in $CO_2RR$ was examined. The $TeN_2$–$CuN_3$ DAC catalyst delivered a stable current density and maintained a high $FE_{CO}$ of > 90%. In addition, the operando XAFS spectra of $TeN_2$–$CuN_3$ DAC and $CuN_4$ SAC at Cu K-edge were conducted using a customized H-type electrolytic cell. The absorption edges in both catalysts shift to lower energy at more negative potentials, and are almost recovered after the applied potential was removed (Supplementary Figs. 25 and 26), indicating the structures of $TeN_2$–$CuN_3$ and $CuN_4$ are stable during $CO_2RR$. Furthermore, the morphology of $TeN_2$–$CuN_3$ DAC after $CO_2RR$ was re-examined by HAADF-STEM. The results indicate that Te and Cu sites are still atomically dispersed on the support, and ~80% of the spots are dual-atom sites (Supplementary Fig. 27), close to that of original $TeN_2$–$CuN_3$ DAC, further demonstrating the $TeN_2$–$CuN_3$ DAC is stable during the $CO_2RR$ process.

## Theoretical modeling

To understand the origins of the higher $CO_2RR$ performance for the $TeN_2$–$CuN_3$ catalyst, density functional theory (DFT) calculations were carried out[37]. To unveil the $CO_2RR$ mechanism, the energy profiles of the $CO_2RR$ on the $TeN_2$–$CuN_3$, $TeN_3$, and $CuN_4$ catalysts without applying electrode potential were investigated[38]. Figures 4a–c show the different atomic structures (upper panels) and corresponding charge difference density plots (bottom panels) for $TeN_2$–$CuN_3$, $TeN_3$ and $CuN_4$. Figures 4d–e show the energy barriers for the transition state of each reaction step in $CO_2RR$. For the step from $CO_2$ to COOH*,

the energy barriers are 1.12 eV (for $TeN_2$–$CuN_3$), 1.72 eV (for $TeN_3$), and 2.02 eV (for $CuN_4$). For the step from COOH* to CO, the energy barriers are 1.18 eV (for $TeN_2$–$CuN_3$), 2.12 eV (for $TeN_3$), and 2.60 eV (for $CuN_4$). Clearly, the $TeN_2$–$CuN_3$ DAC features the lowest energy barrier for the second step. In addition, water dissociation plays a crucial role (by providing protons for $CO_2RR$). Figure 4f shows that during the water dissociation process, the energy barriers for the step from $H_2O$ to H* + OH* are 1.65 eV (for $TeN_3$) and 1.15 eV (for $CuN_4$), implying that the $CuN_4$ can efficiently help to dissociate $H_2O$. For $TeN_2$–$CuN_3$ DAC, the energy barrier is the lowest (0.72 eV). The above data reveal that DAC can simultaneously decrease the energy barriers for $CO_2$ to COOH*, COOH* to CO, and $H_2O$ to H*, which can synergistically catalyze the entire process for $CO_2RR$. Figure 4g shows the calculated configurations for the conversions of $CO_2 \rightarrow$ COOH* and COOH* $\rightarrow$ CO over $TeN_2$–$CuN_3$ DAC. The simulated $TeN_2$–$CuN_3$ site (featuring a distorted, asymmetric structure) could effectively lower the reaction barriers for $CO_2RR$ and water dissociation, and therefore shows the highest $FE_{CO}$ and $j_{CO}$. The steps of transitions from $CO_2$ to COOH*, COOH* to CO, and $H_2O$ to H* and OH* for both $TeN_3$ and $CuN_4$ SACs were found (Supplementary Figs. 28 and 29). From both the experimental data (Fig. 3) and theoretical calculations (Fig. 4d–f), it is noticeable that for $CO_2$ activation, the Te-containing samples outperform $CuN_4$ (without Te), and for water dissociation, the Cu-containing samples outperform $TeN_3$ (without Cu); these results suggest that Te and Cu may play different roles in the DAC for $CO_2RR$. In this regard, we conducted infrared spectroscopy for further verification.

## In situ ATR-SEIRAS

We carried out in situ ATR-SEIRAS to investigate the behavior of Cu and Te species during $CO_2RR$ (Supplementary Fig. 30). The tests were conducted under equilibrium adsorption of $CO_2$. The adsorbed CO can be used as a probe molecule for $CO_2RR$. The peak located in the 1840-1989 cm$^{-1}$ region is attributed to CO adsorption for the sample of $TeN_2$–$CuN_3$ (Fig. 5a)[39]. As the potential becomes more positive, the CO adsorption band shifts to higher wavenumbers, which is due primarily to the vibrational Stark effect[40,41]. Similar trends were also observed for $TeN_3$ (Fig. 5b) and $CuN_4$ (Fig. 5c) SACs, as well as the NC support (Fig. 5d). We noticed that the behavior of CO adsorption on $TeN_2$–$CuN_3$ DAC is almost identical to that on $TeN_3$ in terms of both peak position and Stark tuning rate, but different from those on $CuN_4$ and NC support (as summarized in Fig. 5e), indicating that CO molecules prefer to adsorb on the Te center in the $TeN_2$–$CuN_3$ DAC[42,43]. Therefore, it is highly likely that $CO_2$ molecules are adsorbed on the Te centers and then reduced into adsorbed CO species during $CO_2RR$.

The broad peak from 3450 to 3480 cm$^{-1}$ in Fig. 5a corresponds to the water stretching mode on $TeN_2$–$CuN_3$ DAC, which shifts to higher wavenumbers as the potential goes more positive. The water stretching peaks for $TeN_3$ and $CuN_4$ locate in the regions of 3320−3404 cm$^{-1}$ and 3442−3467 cm$^{-1}$, respectively[44]. Notably, the water stretching peaks for the $TeN_2$–$CuN_3$ and $CuN_4$ locate approximately in the same region, suggesting that water prefers to bind on the Cu centers during $CO_2RR$. The Stark tuning rates of water stretching mode in $TeN_2$–$CuN_3$, $TeN_3$, $CuN_4$, and NC catalysts are 18.2, 51.0, 14.8, and 0 cm$^{-1}$ V$^{-1}$, respectively (Fig. 5f). Both the peak position and Stark tuning rate of the water stretching mode for $TeN_2$–$CuN_3$ are quite similar to those for $CuN_4$, but different from those for $TeN_3$. Furthermore, a peak emerges ~2000 cm$^{-1}$ in the potential range of 0–0.4 V for $TeN_2$–$CuN_3$ and $CuN_4$[44–46], but not for $TeN_3$ or CN support; this peak is attributed to the adsorbed H on the Cu centers, and thus further confirms that water molecules prefer to adsorb onto the Cu centers.

On the basis of the DFT and ATR-SEIRAS results, we can now depict the three critical catalytic steps as shown in Figs. 4 and 5. In the $TeN_2$–$CuN_3$ DAC, the Te center activates $CO_2$, while the Cu center catalyzes $H_2O$ dissociation, yielding protons for further protonation, promoting the $CO_2RR$ process via a synergistic mechanism.

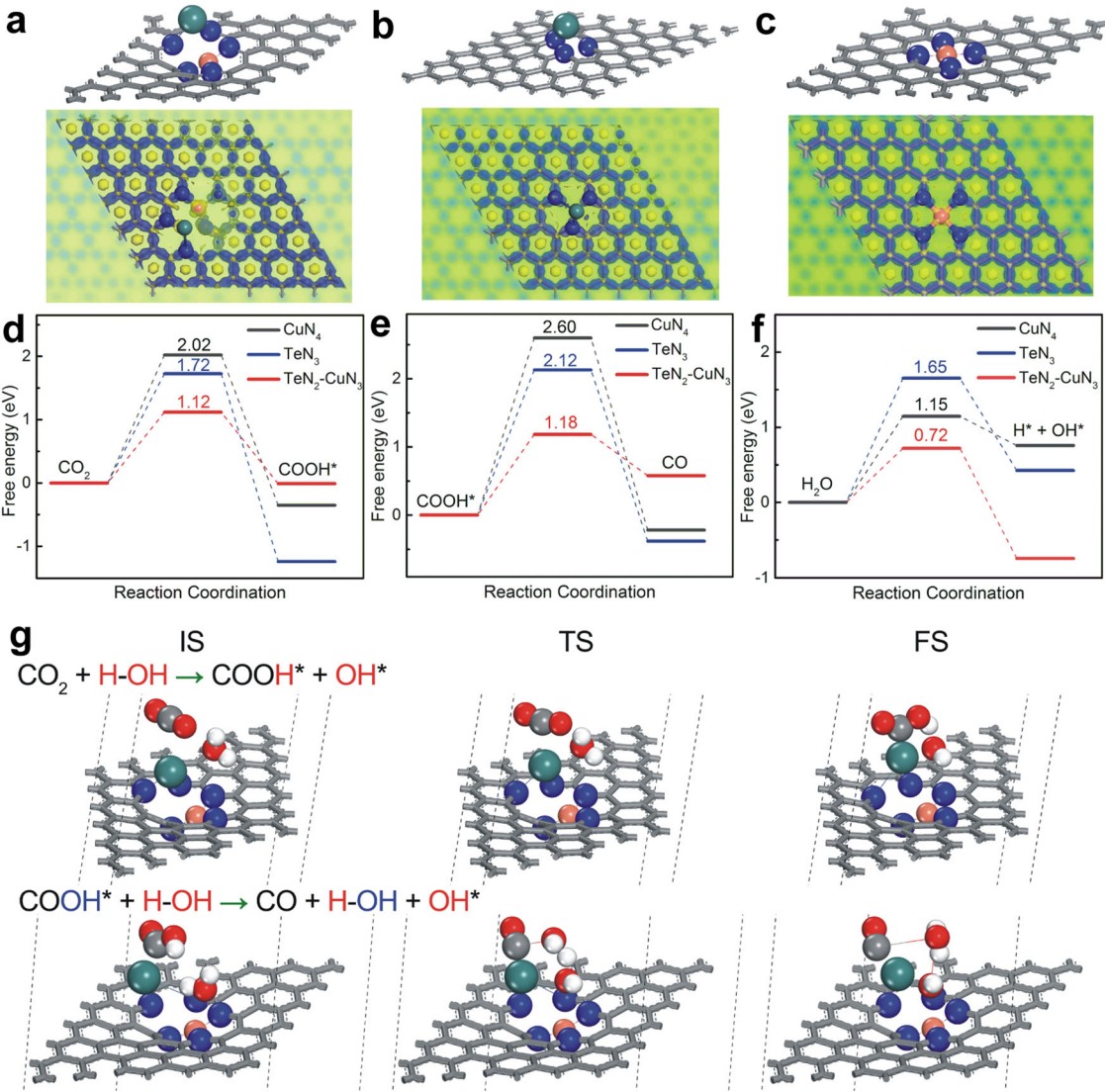

**Fig. 4 | The DFT-based energy barriers of the optimized TeN₃–CuN₃, TeN₃ and CuN₄ catalyst models.** The differential of atomic structure (upper) and corresponding electron density (bottom) for TeN₂–CuN₃ (**a**), TeN₃ (**b**), and CuN₄ (**c**) catalysts; **d** from the CO₂ transition to COOH*, **e** from the COOH* transition to CO during CO₂RR, and **f** from the H₂O transition to H* and OH* during the water dissociation process. **g** Calculated configurations for the conversions of CO₂→COOH* and COOH*→CO (IS, initial state; TS, transition state; FS, final state) over TeN₂–CuN₃ DAC.

## Discussion

We constructed a TeN₂–CuN₃ DAC and revealed the synergistic mechanism for improving CO₂RR. The structural features of TeN₂–CuN₃ DAC were characterized via both XAFS and PDF analyses. For CO₂RR, the DAC displays a broad potential range with high FE$_{CO}$ (>90%), a small Tafel slope (65.2 mV dec⁻¹), and good stability (over 96 h). DFT calculations unveil that the TeN₂–CuN₃ sites could synergistically lower the energy barriers for the CO₂-to-COOH* step and H₂O dissociation, thus effectively promoting the proton transfer kinetics. In situ ATR-SEIRAS gave direct spectroscopic evidence that the TeN₂–CuN₃ site boosts the CO₂RR via a synergistic mechanism: the Te center activates CO₂, and the Cu center helps to dissociate H₂O.

In comparison with previously reported DACs, the main advantages of TeN₂–CuN₃ DAC are as following: (1) Most of the reported DACs employ transition metals as the active centers, which usually display symmetric or quasi-symmetric configurations. In this work, we selected a semimetal (Te) and a transition metal (Cu) to construct the DAC. The atomic numbers of Te and Cu are rather different, and thus leading to an asymmetric structure of TeN₂–CuN₃ owing to different atom sizes, electron configurations and coordinating abilities. (2) The

rather different atomic numbers of Te and Cu in TeN₂–CuN₃ is beneficial for structural characterization by HAADF-STEM, as the brightness of Te and Cu is much different (Fig. 1g). (3) The asymmetric double-atomic sites in TeN₂–CuN₃ DAC can let us to use ATR-SEIRAS to identify the rules of Te and Cu during the electrocatalytic CO₂ reduction, in which Te site mainly activates CO₂, and Cu sites helps to dissociate H₂O, thus provide the direct spectroscopic evidence for synergistic mechanism. We believe that the synthesis strategy and the synergistic mechanism reported here can be of guidance for preparing advanced catalysts with multiple active centers for heterogeneous catalysis.

## Methods
### Chemicals

Na₂TeO₃, poly(vinylpyrrolidone) (PVP, K30), hydrazine hydrate (50% w/w%), copper acetate, Zn(NO₃)₂, methanol, ethanol, aqueous ammonia solution (25−28% w/w%), hydrazine hydrate, and isopropyl alcohol were purchased from Shanghai Chemical Reagent Co. Ltd. Nafion (5 wt% in mixture of lower aliphatic alcohols and water, contains 45% water) was purchased from Sigma-Aldrich. The carbon paper sigracet39BC was purchased from FuelCellsEtc. Ag/AgCl reference

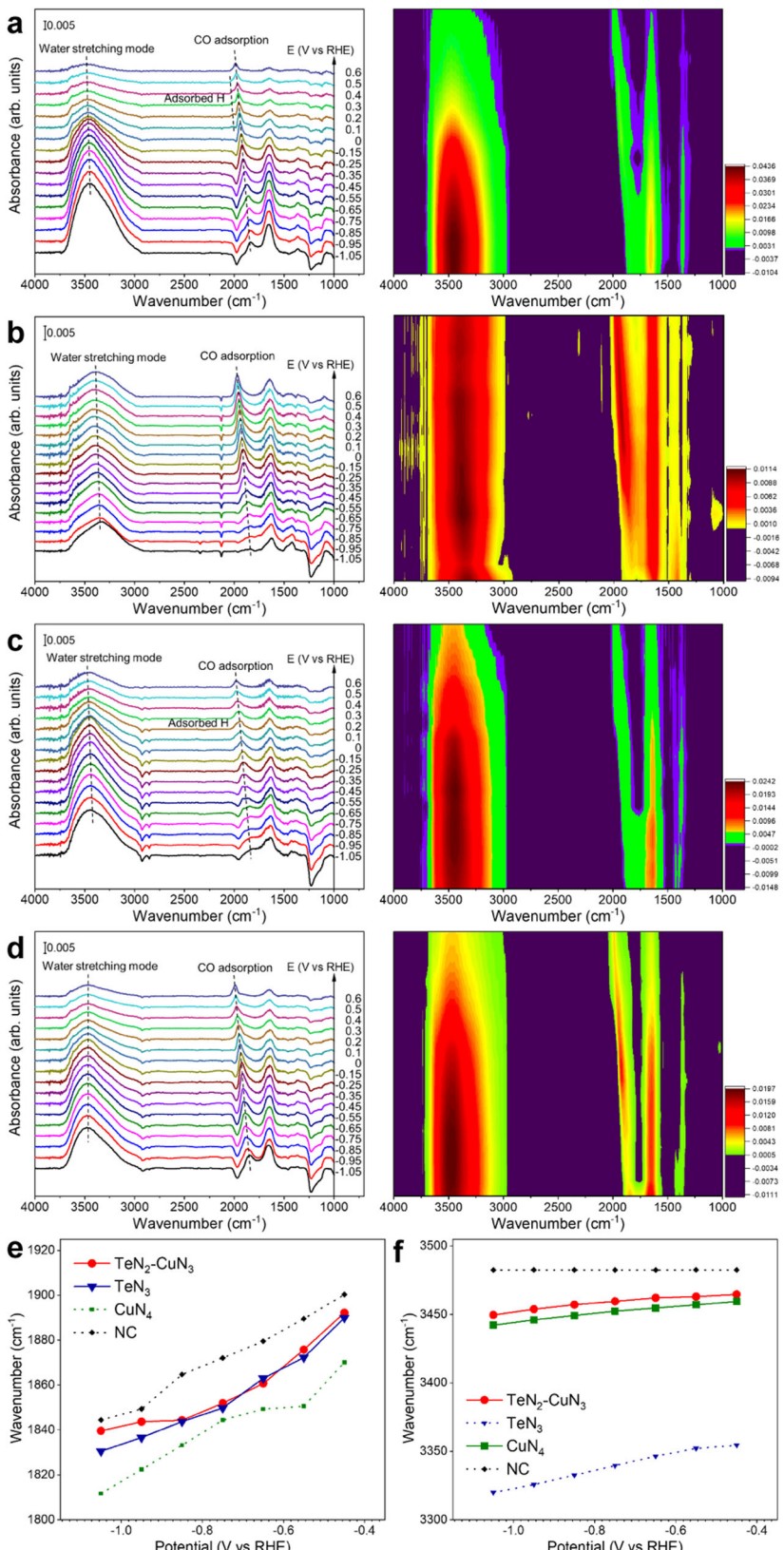

**Fig. 5 | In-situ ATR-SEIRAS analysis of the catalysts.** Spectral profiles for TeN$_2$–CuN$_3$ (**a**), TeN$_3$ (**b**), CuN$_4$ (**c**) and CN (**d**) catalysts in CO$_2$-saturated KHCO$_3$ solution. The applied reference spectral profile was collected at open-circuit potential (OCP). **e**, **f** Plots of potential against wavenumbers in the CO adsorption (**e**) and water stretching mode (**f**) in ATR-SEIRAS spectra.

electrode and Pt plate electrode were purchased from Gaoss Union. All the chemical reagents were used as received without further purification. Sodium carbonate (99.999%) was purchased from Acros. High-purity carbon dioxide gas (99.999%) and nitrogen gas (99.9999%) were purchased from Praxair. Nafion 117 membrane was purchased from DuPont. 18.2 MΩ cm ultrapure water was obtained from milli-Q integral system, and was used for all preparations in the synthesis and $CO_2$ electroreduction tests. All chemical reagents involved were of analytical grade, and used without any further purification. All aqueous solutions were prepared using ultrapure water (DIW, 18.2 MΩ cm).

## Preparation of the DACs and SACs

**Synthesis of Te nanowires (NWs).** Te nanowires were fabricated according to a previously reported method. In brief, PVP (2.0 g) and 177.3 mg of sodium tellurite were dissolved in 70 mL DIW under stirring. Subsequently, ammonia solution (25%, 6.7 mL) and hydrazine hydrate (85%, 3.5 mL) were sequentially added into the above solution. After vigorous magnetic stirring at room temperature for 30 min, the final solution was transferred into a 100 mL Teflon-lined stainless steel autoclave and heated at 180 °C for 3 h. After naturally cooling down to room temperature, Te NWs were precipitated by adding acetone and then redispersed in 50 mL of methanol.

## Synthesis of core–sheath structured Te NWs@ZIF-8

$Zn(NO_3)_2$ methanol solution (0.1 M, 24 mL) was mixed with the obtained Te NWs methanol solution (15 mL). 2-methylimidazole methanol solution (0.8 M, 24 mL) was added to the above solution under stirring for 2 h at room temperature. Then the sample was centrifuged, and the solid was washed with methanol for three times. Finally, the core-shell structured Te NWs@ZIF-8 was obtained after vacuum drying.

## Synthesis of $TeN_2$–$CuN_3$ DAC

A certain amount of core-shell structured Te NWs@ZIF-8 (85 mg) was dispersed into n-hexane (20 mL) via ultrasonication. 20 μL copper acetate methanol solution (8.5 mg/mL) was added dropwise. The mixture solution was stirred under room temperature for 3 h. Subsequently, the sample was centrifuged, and the solid was washed with n-hexane and dried in vacuum. The precursor was heated to 1000 °C for 3 h at a ramping rate of 5 °C/min in $N_2$ flow. The double-atomic sites $TeN_2$–$CuN_3$ was obtained finally. The catalysts with different Cu loadings could be obtained by changing the amount of copper acetate methanol solution (10 μL, 20 μL, 40 μL, 80 μL, and 200 μL).

## Synthesis of TeN3 SACs, CuN4 SACs, and CN

$TeN_3$ SAS: The prepared core-shell structured Te NWs@ZIF-8 was annealed at 1000 °C for 3 h at a ramping rate of 5 °C/min in $N_2$ flow. The $TeN_3$ SAS was thus obtained, without adding Cu.

$CuN_4$ SAS: a certain amount of ZIF-8 (55 mg) was dispersed into n-hexane via ultrasonication. Copper acetate methanol solution (8.5 mg/mL) was added dropwise. The sample was annealed at 1000 °C for 3 h at a ramping rate of 5 °C/min in $N_2$ flow.

CN: The ZIF-8 was annealed at 1000 °C for 3 h at a ramping rate of 5 °C/min in $N_2$ flow.

## Characterization

Power XRD patterns were obtained using a Smart X-ray diffractometer (SmartLab 9 KW, Rigaku, Japan) with Cu Kα radiation ($\lambda = 1.54178$ Å). XPS were taken on ESCA LAB 250 Xi. X-ray photoelectron spectrometer with the Al Kα radiation as the excitation source. The TEM images were obtained carried out using a Tecnai G2 Spirit TWIN. The HAADF-STEM images were taken using a Transmission Electron Microscope with A Probe Corrector (Titan Themis Cubed G2 60-300, FEI). The BET (Brunauer−Emmett−Teller) test was obtained from multi-station specific surface micropore and vapor adsorption analyzer (BELSORP-Mas, MicrotracBEL, Japan). The loadings of Te and Cu on

carbon cloth were determined via inductively coupled plasma-atomic emission spectroscopy (ICP-AES, SPECTRO-BLUE).

## Electrochemical measurements

All electrochemical measurements were carried out in a three-electrode system on a CHI 760e workstation (Shanghai CHI Instruments Company) at 25 °C. A gas-tight H-type cell with two compartments separated by a proton exchange membrane (Nafion 117, DuPont) was used in the electrochemical tests. The prepared catalysts were used directly as the working electrodes, and Pt plate and Ag/AgCl (saturated KCl) were used as the counter and reference electrode, respectively. The electrolyte was bubbled with $CO_2$ or Ar for at least 30 min to form $CO_2$-saturated solution and maintained the flow rate of 20 sccm during measurements. The 2.5 mg catalysts and 20 μL Nafion solution were ultrasonically mixed with 200 μL of 2-propanol and 800 μL ultrapure water to form a homogeneous ink. The working electrode was obtained by dispensing the catalyst ink on the surface of carbon cloth (electrode surface area: $0.5 \times 0.5$ cm$^2$, catalyst loading, 0.5 mg cm$^{-2}$). The EIS measurements were carried out with 100 mV amplitude in a frequency range from $10^5$ Hz to 0.1 Hz.

The flow cell consists of a gas chamber, a catholyte chamber and an anolyte chamber. Each chamber has an inlet and outlet for the feed of $CO_2$ gas or for the circulation of electrolyte. The exposed window for electrode is $1 \times 1$ cm$^2$. 1 M KOH aqueous solution was used as both anolyte and catholyte and the two chambers were separated with anion exchange membrane. An electronic flowmeter was employed to control the flow rate of $CO_2$ gas.

All potentials were converted to the RHE scale using the Nernst equation:

$$E(vs.\,\mathrm{RHE}) = E(vs.\,\mathrm{Ag/AgCl}) + 0.197V + 0.059 \times pH \quad (1s)$$

The Faradaic efficiencies (FEs) for CO and $H_2$ were calculated according to the following equation:

$$FE = \frac{z.n.F}{Q} \quad (2)$$

FE is faradaic efficiency for CO or $H_2$; z is the number of electrons transferred to the product; n is the amount of substance of the product; F is Faraday constant (96485 C/mol); Q is the input charge (C).

CO partial current density ($j_{CO}$) can be obtained according to the following equation:

$$J_{co} = \frac{FE_{CO} \cdot I}{S} \quad (3)$$

where I is the average current, and S represents the geometric surface area of the working electrode.

## XAFS measurements

For the $TeN_2$–$CuN_3$ DAC, the XAFS spectra at Te K-edge were collected at BL14W1 beamline of Shanghai Synchrotron Radiation Facility (SSRF), and the data at Cu K-edge were collected at 1W1B beamline of Beijing Synchrotron Radiation Facility (BSRF) and BL11B beamline of SSRF. All the data above were collected in fluorescence mode using a Lytle detector. For the corresponding oxide reference samples, the XAFS data were collected in transmission mode on TableXAFS-500A at Anhui Chuangpu Instrument Technology Co., Ltd. All the samples were ground and uniformly daubed on the special adhesive tape. The acquired EXAFS data were processed according to the standard procedures using the ATHENA software of Demeter software packages[47]. The EXAFS spectra were obtained by subtracting the post-edge background from the overall absorption profile and then normalizing with respect to the edge-jump step. The $R_{bkg}$ value equaled to 1.1 for all

samples. Subsequently, the χ(k) data were Fourier transformed to (R) space using a hanning window ($dk = 1.0\,\text{Å}^{-1}$) to separate the EXAFS contributions from different coordination shells. In order to derive the quantitative structural parameters around specific central atoms, least-squares curve parameter fitting was carried out using the ARTE-MIS software of Demeter software packages.

The following EXAFS equation was used to calculate the theoretical scattering amplitudes, the phase shifts, and the photoelectron mean free paths for all paths:

$$\chi(k) = \sum_j \frac{N_j S_0^2 F_j(k)}{kR_j^2} \cdot \exp\left[-2k^2\sigma_j^2\right] \cdot \exp\left[\frac{-2R_j}{\lambda(k)}\right] \cdot \sin\left[2kR_j + \phi_j(k)\right] \quad (4)$$

$S_0^2$ − amplitude reduction factor; $F_j(k)$ − effective curved-wave backscattering amplitude; $N_j$ −number of neighbors in the $j^{th}$ atomic shell; $R_j$ − distance between the X-ray absorbing central atom and the atoms in the $j$th atomic shell (back scatterer); $\lambda$ − mean free path in Å; $\phi_j(k)$ − the phase shift (including the phase shift for each shell and the total central atom phase shift); $\sigma_j$ − Debye−Waller parameter of the $j^{th}$ atomic shell (variation of distances around the average $R_j$).

The functions $F_j(k)$, $\lambda$ and $\phi_j(k)$ were calculated using the ab initio code FEFF10. The additional details for EXAFS fitting are given below.

All fittings were performed in the $R$ space with a $k$-weight of 2, and phase correction was applied in the first coordination shell to render the $R$ values close to the physical interatomic distances between the absorbers and shell scatterers. The CNs for model samples were fixed as the nominal values. The amplitude reduction factor $S_0^2$, the internal atomic distances $R$, the Debye−Waller factor $\sigma2$, and the edge-energy shift Δ were allowed to run freely. For $TeN_2$−$CuN_3$, the EXAFS spectral fitting was conducted under the boundary conditions for Te K-edge with $k$ ranging from 2.771 to 13.196, R ranging from 1.35 to 4.00 employing seven variables in XAFS fitting table and 14.27 independent points; for Cu K-edge, $k$ ranges from 2.642 to 14.041, R ranges from 1.3 to 4.00 employing seven variables and 15.97 independent points. For $TeN_3$, the EXAFS spectral fitting was conducted under the boundary conditions for Te K-edge with $k$ ranging from 2.739 to 10.797, R ranging from 1.30 to 4.00 employing seven variables and 13.34 independent points. For $CuN_4$, the EXAFS spectral fitting was conducted under the boundary conditions for Cu K-edge with $k$ ranging from 2.411 to 12.702, R ranging from 1.3 to 4.0 employing seven variables and 12.87 independent points.

All fits were performed in the $R$ space with $k$-weight of 2, and phase correction was also applied in the first coordination shell to make $R$ value close to the physical interatomic distance between the absorber and shell scatterer. The CNs of model samples were fixed as the nominal values. The $S_0^2$, internal atomic distances $R$, Debye−Waller factor $\sigma^2$, and the edge-energy shift Δ were allowed to run freely.

The X-ray total scattering data were collected at BL17b beamline in energy state of 20 keV (0.6199 Å) of National Facility for Protein Science (NFPS) of SSRF. The 2D XRD patten was first integrated to obtain 1D total scattering intensity $I(Q)$ (calibrated by $CeO_2$ celebrant by Dioptas0.5.2 package[48]. Additional scattering measurements from kapton capillary were performed under the same conditions for background subtraction. Then the reduced pair distribution function $G(r)$ was obtained through Fourier-transform total scattering structure function $S(Q)$ derived from $I(Q)$ by PDFgetX3[49].

The following $G(r)$ equation was used:

$$G(r) = \frac{2}{\pi} \int_0^\infty Q[S(Q) - 1]\sin(Qr)dQ \quad (5)$$

## Operando XAFS characterization

The operando XAFS spectroscopy was conducted using customized two-compartment H-type electrochemical cell. The cell involved the working, counter (Pt wire), and references electrodes (Ag/AgCl) as well as feed gas ($CO_2$, 99.999%), inlets, and outlets. The prepared catalyst was drop-cast on a carbon paper (SigracetGDL29BC) electrode and dried overnight. A small window was cut out on the cathode side and sealed with Kapton film to allow fluorescence signals to pass from the electrode to the detector. To record the operando XANES spectra, the cell was subsequently filled with electrolyte. 0.1 M $KHCO_3$ aqueous solution, and bubbled with $CO_2$ for 30 min before test. The XANES spectra were collected under operando conditions at open-circuit potential (OCP) and different applied potentials.

## In situ attenuated total reflection surface-enhanced infrared absorption spectroscopy (ATR-SEIRAS)

*Materials:* Sodium hydroxide (NaOH, 97%), ammonium chloride ($NH_4Cl$, 99.5%), sodium sulfite ($Na_2SO_3$, 98%), sodium thiosulfate pentahydrate ($Na_2S_2O_3\,5H_2O$, 99.99%) and hydrofluoric acid (HF, 40%) were purchased from Aladdin. Nitric acid ($HNO_3$, 65%), hydrochloric acid (HCl, 36%), potassium bicarbonate ($KHCO_3$, 99.5%), and ethanol (99.7%) were purchased from Sinopharm. Sodium tetra-chloroaurate(III) dihydrate ($NaAuCl_4\,2H_2O$, 98%) was purchased from Bidepharm. Nafion solution and ammonium fluoride ($NH_4F$, 98%) was purchased from Sigma-Aldrich. Ag/AgCl reference electrode and Pt wire electrode were purchased from CH Instruments. Deionized water (DIW) was obtained from a distillation apparatus. All the electrochemical experiments were conducted with a VersaSTAT 3 F electrochemical workstation and the SEIRAS spectra were collected by Nicolet iS50 FT-IR.

## Preparation of Au plating solution

In a typical process, 0.1143 g $NaAuCl_4\,2H_2O$ was dissolved in 1.5 mL DIW and followed by adding 0.0611 g NaOH. The solution turned from a transparent yellowish color to translucent orange upon addition of NaOH. Then 0.067 g $NH_4Cl$, 0.4734 g $Na_2SO_3$, and 0.3101 g $Na_2S_2O_3\,5H_2O$ were dissolved in 25 mL water. The two solution were mixed in a volumetric flask and 25 mL of DIW was added into it. Finally, the solution was sonicated for 2 h and sat overnight for further use.

## Au film deposition

A Si crystal was first immersed in aqua regia solution to remove previous film, then polished with 0.05 μm $Al_2O_3$ powder until the surface became hydrophobic. The $Al_2O_3$ powder was washed off the surface of the crystals by sonicating in water bath. Next, the cleaned crystal was immersed in a 40% $NH_4F$ bath for 2 min to form a hydride-terminated surface. Then the crystal was immersed in the Au seeding solution containing 3.75 mL Au plating solution and 0.86 mL 2% HF, under 55 °C for 4−5 min. Finally, the crystal was rinsed with DIW.

## Au film activation and electrochemical measurement

All electrochemical experiments were performed in a custom-made three-electrode cell, including an Au film working electrode, an Ag/AgCl reference electrode, and a Pt wire counter electrode. The Au films were activated via cyclic voltammetry for 10 cycles between −0.2 V and 1 V vs. RHE with a scan rate of 50 mV s⁻¹ in order to improve the signal. Then dispersed materials in 2% Nafion solution and modified Au film with these suspensions. The $CO_2$ reduction reaction was conducted in 0.5 M $KHCO_3$ via chronoamperometry with continuous purge of $CO_2$, and the potential range was from −1.05 V to 0.6 V vs. RHE. All spectra were collected at a 4 cm⁻¹ spectral resolution.

## Density functional theory (DFT) calculations

All DFT calculations were performed using the plane-wave pseudo-potential method, with the CASTEP module implemented in Material Studio 6.0. The Generalized Gradient Approximation with Perdew−Burke−Ernzerhof exchange-correlation functional was used to describe the exchange-correlation effects. We used plane-wave basis

with a cut-off energy of 400 eV, self-consistent field tolerance of $1 \times 10^{-5}$ eV, maximum force 0.1 eV per Å, and maximum displacement 0.005 Å respectively for the geometry optimization.

The pathway of $CO_2RR$ reactions in weak alkaline electrolyte:

1. $CO_2 + H^* \rightarrow COOH^*$
2. $COOH^* + H^* \rightarrow CO + H_2O$
3. $H_2O \rightarrow H^* + OH^*$

## Data availability

All data generated or analyzed during this study are included in the published article and its supplementary information files.

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

## Acknowledgements

This work was financially supported by the National Natural Science Foundation of China (No.52072260, No.21931007, No.21790052 and No.51403114), Science and Technology Support Program for Youth Innovation in Universities of Shangdong Province (No.2020KJA012), Tianjin Natural Science Foundation (21JCZXJC00130, B2021201074), National Key R&D Program of China (No.2017YFA0700104), Haihe Laboratory of Sustainable Chemical Transformations, Dalian high-level talent innovation project (No.2019RQ063), HUST Academic Frontier Youth Team grant (Grant No. 2019QYTD11). We acknowledge BL14W1 and BL11B beamlines of Shanghai Synchrotron Radiation Facility (SSRF) (Shanghai), and 1W1B beamline of BSRF (Beijing) for providing the beam time.

## Author contributions

J.J. and T.L. conceived the project. M.T., X.H., and Z.S. carried out the syntheses, structural characterizations, and CO2RR test. X.Y. and Q.Y. carried out the situ ATR-SEIRAS experiment and provided the analysis, M.G. and H.X. provided theoretical analysis, and J.Z. provided the test of the XANES and EXAFS. J.J., Z.S., and Y.M. carried out the operando XAFS spectroscopy. J.J. and T.L. analyzed all the data and prepared the manuscript. C.Z. helped with polishing this manuscript. All of the authors participated in preparing the manuscript.

## Competing interests

The authors declare no competing interests.
