## [Peer Review File · Nature Communications]

REVIEWER COMMENTS

Reviewer #1 (Remarks to the Author):

Jiao et al. constructed an asymmetric TeN₂-CuN₃ DAC for CO₂RR, and revealed a “diatomic-activating-bimolecular” catalytic mechanism. The asymmetric structure of TeN₂-CuN₃ DAC site was characterized by combining data from full-range synchrotron pair distribution function (PDF) and synchrotron-radiation-based X-ray absorption spectroscopy (XAS). For CO₂RR, the DAC displays a broad potential range of high FECO and a good stability in both H-type cell and flow cell. In situ ATR-SEIRAS gave direct spectroscopic evidence for a “diatomic-activating-bimolecular” synergistic mechanism. I think the synthesis scheme and the characterization methodology would be of interest for relevant researchers, and this work is suitable for publication in Nature Communications after minor revision.

1. The DAC has a larger BET surface area than the reference samples. The authors need to confirm that the elevated performance come from the boosting effect of the formed Te-Cu DAC rather than the large surface area.
2. Regarding Figure 2 XAFS and PDF analysis for TeN₂-CuN₃ DAC. “the radial distance space spectra $\chi(R)$ for the TeN₃ and TeN₂-CuN₃ show a major peak at ~ 1.85 Å and 1.91 Å, respectively, which is attributed to the Te-N scattering path.” Are there any Te-Te bonds or Cu-Cu bonds in the DAC catalyst? Please clarify.
3. In DFT calculations, the TeN₃-CuN₃ has a symmetric structure. Why would TeN₂-CuN₃ form rather than TeN₃-CuN₃?
4. As the FECO (upper panel) and FEH₂ (bottom panel) for the samples TeN₂-CuN₃, TeN₃, CuN₄ and NC were evaluated for different cathodic potentials, why would the performance decline at higher Cu loadings for the Te-Cu catalyst?
5. In ATR-SEIRAS, why was the signal for adsorbed CO₂ signal not observed?

Reviewer #2 (Remarks to the Author):

This paper reports the construction of an asymmetric TeN₂/CuN₃ diatomic sites and the catalytic performance for CO₂ electroreduction. The fixation method of the asymmetric Te/Cu structure is noteworthy, and the detailed full range synchrotron spectrum has been analyzed very carefully. Combining operando IR spectrum with theoretical calculations, a feasible CO₂RR pathway that the diatomic centers lower the energy barriers to promote proton-transfer kinetics was concluded. Overall,

this paper was organized well and data are consistent to support the main viewpoint. It can be accepted after the concerns are fully addressed.

1. The author selected the pair of Te and Cu. What is the particular advantage of this combination in comparison with other diatomic sites?
2. The formation of the DAC during the synthesis process needs to be explained in further depth.
3. How was the TeN₂-CuN₃ structure identified?
4. Why would higher Cu loadings lead to the decline in the activity and selectivity for the DAC?
5. It is said that CO₂ is activated via a "diatomic-activating-bimolecular" mechanism. However, no signals corresponding to CO₂ were found in the ATR-SEIRAS results. Could the authors add some discussion and explanation on this matter?

Reviewer #3 (Remarks to the Author):

The paper reports the synthesis and characterization of "an asymmetric TeN₂-CuN₃ double-atomic site catalyst (DAC)" for CO₂ reduction reactions (CO₂RR). Scanning transmission emission microscopy (HAADF-STEM) and synchrotron-based XAFS were respectively used to examine the atomic dispersion and coordination of Te and Cu in DAC and in reference compounds with "single-atomic" sites. These characterization methods are, however, performed *ex situ*, i.e. under conditions far removed from the operating conditions of the electrocatalyst. The presence and stability of these tantalizing "single-" and "double-atomic" sites are mostly unverified in electrochemical environments. Surface reconstruction can transpire after these DACs are loaded onto an electrode in contact with water, CO₂, and supporting electrolytes and then subsequently subjected to very negative potentials. A typical, albeit inadequate, remediation to this predicament is the pre-catalysis post-catalysis approach to structural characterization, which is crucially missing in this paper. In the absence of operando surface structural information, the reported CO₂RR performances of TeN₂-CuN₃, TeN₃, CuN₄, CN, *inter alia*, cannot be directly ascribed to specific surface-atom sites or structures. The persistence of these surface sites at the intentionally roughened ATR-SEIRAS substrate has to be unambiguously established before the term operando can be attached to the spectroscopic technique. Before the paper can be considered for publication, it is strongly recommended that an experimental proof be provided for the presence of the asymmetric double-atomic site under actual CO₂RR potentials.

Point-by-point response to the reviewers' comments

Reviewer #1

Jiao et al. constructed an asymmetric TeN₂-CuN₃ DAC for CO₂RR, and revealed a “diatomic-activating-bimolecular” catalytic mechanism. The asymmetric structure of TeN₂-CuN₃ DAC site was characterized by combining data from full-range synchrotron pair distribution function (PDF) and synchrotron-radiation-based X-ray absorption spectroscopy (XAS). For CO₂RR, the DAC displays a broad potential range of high F_{Eco} and a good stability in both H-type cell and flow cell. In situ ATR-SEIRAS gave direct spectroscopic evidence for a “diatomic-activating-bimolecular” synergistic mechanism. I think the synthesis scheme and the characterization methodology would be of interest for relevant researchers, and this work is suitable for publication in Nature Communications after minor revision.

We much appreciate the Reviewer's positive comments on the significance of our work. We have made changes according to the Reviewer's constructive suggestions. Detailed responses are provided in the following.

1. The DAC has a larger BET surface area than the reference samples. The authors need to confirm that the elevated performance come from the boosting effect of the formed Te-Cu DAC rather than the large surface area.

Reply: We thank the reviewer's thoughtful comments. Compared with BET surface area, the electrochemically active surface area (ECSA) could more properly assess the surface area for electrochemical CO₂RR. In order to determine the surface area for CO₂RR, the ECSA was measured using the double-layer capacitance (C_{dl}) method. The C_{dl} of TeN₂-CuN₃, TeN₃ and CuN₄ catalysts are 112, 99 and 88 mF cm⁻², respectively (Supplementary Fig. 21). The ECSA of TeN₂-CuN₃, TeN₃ and CuN₄ catalysts are deduced as 700.4, 676.2 and 552.1 cm², respectively. The C_{dl} and ECSA

of TeN₂-CuN₃ are only slightly larger than those of TeN₃ and CuN₄, while the j_{CO} of TeN₂-CuN₃ is about 4 and 8 times higher than those of TeN₃ and CuN₄ (at -0.65 V), respectively. Furthermore, the TeN₂-CuN₃ displays a high FE_{CO} (> 90%) over a broad potential window from -0.55 V to -0.85 V. By contrast, the FE_{CO} for TeN₃ reaches a maximum of 80% at -0.65 V, and the FE_{CO} for CuN₄ reaches the maximum of 60% at a more negative potential (-0.85 V). Thus, we can conclude that the enhanced CO₂RR activity of TeN₂-CuN₃ is mainly attributed to synergistic effect between Te and Cu active sites, rather than the ECSA.

Supplementary Fig. 21 CV curves at various scan rates (from 2 to 10 mV s⁻¹) (a-c), and the corresponding C_{dl} (d) and ECSA values (e) for TeN₂-CuN₃, TeN₃, and CuN₄.

2. Regarding Figure 2 XAFS and PDF analysis for TeN₂-CuN₃ DAC. “the radial distance space spectra $\chi(R)$ for the TeN₃ and TeN₂-CuN₃ show a major peak at ~1.85 Å and 1.91 Å, respectively, which is attributed to the Te-N scattering path.” Are there any Te-Te bonds or Cu-Cu bonds in the DAC catalyst? Please clarify.

Reply: We thank the reviewer’s insightful question. As shown in Fig. 2b, the radial distance space spectra $\chi(R)$ for the TeN₃ and TeN₂-CuN₃ show a major peak at ~1.85

Å and 1.91 Å, respectively, which is attributed to the Te–N scattering path. Fig. 2b shows that the Te–Te path (2.92 Å) for Te foil. While the Te–N–C scattering paths for TeN₃ and TeN₂–CuN₃ are located at ~2.75 and 2.85 Å, both values are shorter than the Te–Te bond. We conducted data fitting to further distinguish the two paths. As shown in Fig. 2c, the fitting results for Te–N and Te–N–C paths agree well with the experimental data, and the $\chi(R)$ intensity for Te–N–C is far lower than that for Te–Te, which collectively indicate that Te is atomically dispersed on the support, revealing no Te–Te bond in TeN₂–CuN₃ and TeN₃ catalysts. Likewise, in the Cu K-edge $\chi(R)$ space spectra (Fig. 2e), the Cu–N–C paths for CuN₄ and TeN₂–CuN₃ are located at ~2.55 and 2.60 Å, respectively, both larger than the Cu–Cu bond, indicating no Cu–Cu bond in TeN₂–CuN₃ and CuN₄ catalysts. Furthermore, we performed pair distribution function $G(r)$ (PDF ($G(r)$)) analysis. The PDF Rietveld refinement partial result for Cu–Te path in TeN₂–CuN₃ is shown in Fig. 2f. The peak at ~3.29 Å is attributed to Te–Cu path; the other peaks at longer distances are due to the multiple scattering paths (> 5 Å) (Ref. 35: *Science* **2007**, *316*, 561-565; Ref. 36: *Adv. Energy Mater.* **2021**, *11*, 2003304).

Based on the above results, we can depict the structural characteristics of TeN₂–CuN₃ DAC. In TeN₂–CuN₃, there are no Te–Te or Cu–Cu bonds in the DAC catalyst. The distance between Te and Cu atoms is approximately 3.3 Å with no bond formed between Te and Cu, which is consistent with HAADF-STEM image (Fig. 1g) and XAFS analysis (Fig. 2); specifically, the Te and Cu atoms are closely immobilized on the support, coordinated with two and three N atoms, respectively.

Fig. 2 XAFS and PDF analysis for $\text{TeN}_2\text{-CuN}_3$ DAC. **b, e** Radial distance $\chi(R)$ space spectra of Te and Cu. **c** Fitting results for Te-N and Te-N-C paths of $\text{TeN}_2\text{-CuN}_3$ DAC. **f** PDF Rietveld refinement result for partial Cu-Te paths of $\text{TeN}_2\text{-CuN}_3$ DAC.

3. In DFT calculations, the $\text{TeN}_3\text{-CuN}_3$ has a symmetric structure. Why would $\text{TeN}_2\text{-CuN}_3$ form rather than $\text{TeN}_3\text{-CuN}_3$?

Reply: As the reviewer considered, we did construct a $\text{TeN}_3\text{-CuN}_3$ model structure for the optimization in the beginning. The results of DFT analysis revealed that the structure automatically evolves into $\text{TeN}_2\text{-CuN}_3$ after the optimization (Fig. R1). We speculate that this is because the Te atom has a larger size and thus more prominent steric effect, rendering the asymmetric $\text{TeN}_2\text{-CuN}_3$ structure. And this DFT-optimized structure is in good accordance with the EXAFS results.

Fig. R1 TeN₃-CuN₃ structure (a) before and (b) after optimization. Inset: the distances of Te-N and Cu-N.

4. As the F_{ECO} (upper panel) and F_{EH₂} (bottom panel) for the samples TeN₂-CuN₃, TeN₃, CuN₄ and NC were evaluated for different cathodic potentials, why would the performance decline at higher Cu loadings for the Te-Cu catalyst?

Reply: The decline in CO₂ reduction performance at higher Cu loadings for the Te - Cu catalyst is because Cu acts mainly as the catalytic sites for water dissociation, which is favorable for hydrogen evolution reaction. From Fig. 3b, it can be seen that the F_{ECO} for CuN₄ SAC is much lower than that of TeN₂-CuN₃ DAC. Therefore, along with the increase of Cu loading, more Cu single-atom sites are located on the NC support, which leads to the increase of F_{EH₂} and the decrease of F_{ECO}.

5. In ATR-SEIRAS, why was the signal for adsorbed CO₂ signal not observed?

Reply: We thank the reviewer's thoughtful question. The peaks in the ATR-SEIRA spectra reflect the changes relative to the reference spectra. In our study, as the reference spectra were collected in the presence of CO₂-saturated electrolyte at the open-circuit potential (OCP), the change in the adsorbed CO₂ peak in ATR-SEIRA spectra is negligible compared with that in the reference spectra. Indeed, if we take a close look at the SEIRA spectra in the 2300-2400 cm⁻¹ region, there is a tiny peak corresponding to the CO₂ adsorption (Fig. 5a), which is similar with previous reports (ACS Catal., 2018, 8, 3999). The peak is tiny because CO₂ adsorption has reached equilibrium and there is barely change relative to the state at the OCP on the interfaces during the CO₂RR.

Fig. 5 a ATR-SEIRAS spectra of TeN₂-CuN₃.

Reviewer #2

This paper reports the construction of an asymmetric TeN₂/CuN₃ diatomic sites and the catalytic performance for CO₂ electroreduction. The fixation method of the asymmetric Te/Cu structure is noteworthy, and the detailed full range synchrotron spectrum has been analyzed very carefully. Combining operando IR spectrum with theoretical calculations, a feasible CO₂RR pathway that the diatomic centers lower the energy barriers to promote proton-transfer kinetics was concluded. Overall, this paper was organized well and data are consistent to support the main viewpoint. It can be accepted after the concerns are fully addressed.

We much appreciate the reviewer's valuable comments and careful review for our manuscript.

1. The author selected the pair of Te and Cu. What is the particular advantage of this combination in comparison with other diatomic sites?

Reply: In comparison with previously reported DACs, the main advantages of TeN₂-CuN₃ DAC are as following: (1) Most of reported DACs employ transition metals as the active centers, which usually display symmetric or quasi-symmetric configurations. In this work, we selected a semimetal (Te) and a transition metal (Cu) to construct the DAC. The atomic numbers of Te and Cu are rather different, and thus leading to an asymmetric structure of TeN₂-CuN₃ owing to different atom sizes, electron configurations and coordinating abilities. (2) The rather different atomic numbers of Te and Cu in TeN₂-CuN₃ is beneficial for structural characterization by HAADF-STEM, as the brightness of Te and Cu is much different (Fig. 1g). (3) The asymmetric double-atomic sites in TeN₂-CuN₃ DAC can let us to use ATR-SEIRAS to identify the roles of Te and Cu during the electrocatalytic CO₂ reduction, in which Te site mainly activates CO₂, and Cu sites helps to dissociate H₂O, these results can give the direct spectroscopic evidence for “diatomic-activating-bimolecular” synergistic mechanism.

We have added the above statements in the Discussion section (Page 18).

2. The formation of the DAC during the synthesis process needs to be explained in further depth.

Reply: As suggested, the following statements for the formation of TeN₂-CuN₃ DAC during the synthesis process were added in the revised manuscript (Page 4):

“During the pyrolysis process at 1000 °C, the Cu atoms were anchored on the resulting N-doped carbon support. The Te atoms were evaporated from the inside Te nanowires to the outside porous N-doped carbon support. As Cu sites are mainly anchored by the N atoms in N-doped carbon support, thus the subsequently Te-N coordination interaction can let the Te atoms anchored near Cu sites to generate TeN₂-CuN₃ DACs.”

3. How was the TeN₂-CuN₃ structure identified?

Reply: The TeN₂-CuN₃ structure was identified by HAADF-STEM image, XAFS and

pair distribution function (PDF) analysis. From the HAADF-STEM image of TeN₂-CuN₃ DAC, it can be found that the brightness of dual-atom pairs is obviously different (Fig. 1g), demonstrating the formation of heteronuclear TeCu DAC. In addition, the Te-Cu distance of 0.33 nm observed in HAADF-STEM image (Fig. 1g) is consistent with the results of PDF Rietveld refinement (Fig. 2f), further demonstrating the formation of asymmetric TeN₂-CuN₃ DAC.

The results of XAFS fitting (Fig. 2c and Supplementary Fig. 12 and Fig. 14) reveal that the coordination numbers of Te and Cu are 2 and 3, respectively, demonstrating the formation of TeN₂-CuN₃ structure. In addition, the results of DFT analysis revealed that the structure automatically evolves from TeN₃-CuN₃ into TeN₂-CuN₃ after the optimization (please see Reviewer #1, question 3), indicating TeN₂-CuN₃ is more stable than TeN₃-CuN₃.

All the above results confirm the formation of asymmetric TeN₂-CuN₃ structure.

4. Why would higher Cu loadings lead to the decline in the activity and selectivity for the DAC?

Reply: The decline in CO₂ reduction performance at higher Cu loadings for the Te - Cu catalyst is because Cu acts mainly as the catalytic sites for water dissociation, which is favorable for hydrogen evolution reaction. From Fig. 3b, it can be seen that the FE_{CO} for CuN₄ SAC is much lower than that of TeN₂-CuN₃ DAC. Therefore, along with the increase of Cu loading, more Cu single-atom sites are located on the NC support, which leads to the increase of FE_{H₂} and the decrease of FE_{CO}.

5. It is said that CO₂ is activated via a "diatomic-activating-bimolecular" mechanism. However, no signals corresponding to CO₂ were found in the ATR-SEIRAS results. Could the authors add some discussion and explanation on this matter?

Reply: We thank the reviewer's thoughtful question. The peaks in the ATR-SEIRA spectra reflect the changes relative to the reference spectra. In our study, as the reference spectra were collected in the presence of CO₂-saturated electrolyte at the

open-circuit potential (OCP), the change in the adsorbed CO₂ peak in ATR-SEIRA spectra is negligible compared with that in the reference spectra. Indeed, if we take a close look at the SEIRA spectra in the 2300–2400 cm⁻¹ region, there is a tiny peak corresponding to the CO₂ adsorption (Fig. 5a), which is similar with previous reports (ACS Catal., 2018, 8, 3999). The peak is tiny because CO₂ adsorption has reached equilibrium and there is barely change relative to the state at the OCP on the interfaces during the CO₂RR.

Fig. 5 a ATR-SEIRAS spectra of TeN₂-CuN₃.

Reviewer #3

The paper reports the synthesis and characterization of "an asymmetric TeN₂-CuN₃ double-atomic site catalyst (DAC)" for CO₂ reduction reactions (CO₂RR). Scanning transmission emission microscopy (HAADF-STEM) and synchrotron-based XAFS were respectively used to examine the atomic dispersion and coordination of Te and Cu in DAC and in reference compounds with "single-atomic" sites. These characterization methods are, however, performed ex situ, i.e., under conditions far removed from the operating conditions of the electrocatalyst. The presence and stability of these tantalizing "single-" and "double-atomic" sites are mostly unverified

in electrochemical environments. Surface reconstruction can transpire after these DACs are loaded onto an electrode in contact with water, CO₂, and supporting electrolytes and then subsequently subjected to very negative potentials. A typical, albeit inadequate, remediation to this predicament is the pre-catalysis post-catalysis approach to structural characterization, which is crucially missing in this paper. In the absence of operando surface structural information, the reported CO₂RR performances of TeN₂-CuN₃, TeN₃, CuN₄, CN, inter alia, cannot be directly ascribed to specific surface-atom sites or structures. The persistence of these surface sites at the intentionally roughened ATR-SEIRAS substrate has to be unambiguously established before the term operando can be attached to the spectroscopic technique. Before the paper can be considered for publication, it is strongly recommended that an experimental proof be provided for the presence of the asymmetric double-atomic site under actual CO₂RR potentials.

Reply: We much appreciate the reviewer's insightful suggestion. As suggested, we have performed the measurements of operando synchrotron-radiation-based X-ray absorption fine structure (XAFS) spectra, as well as the HAADF-STEM image of TeN₂-CuN₃ DAC after CO₂RR, to identify the stability of TeN₂-CuN₃ DAC under actual CO₂RR potentials.

(1) Operando XAFS measurements (corresponding text has been added to the Supplementary Information, section 1.6, page S7)

The operando XAFS spectroscopy was conducted using customized two-compartment H-type electrochemical cell (Fig. R2). The cell involved the working, counter (Pt wire), and references electrodes (Ag/AgCl) as well as feed gas (CO₂, 99.999%), inlets and outlets. The prepared catalyst was drop-cast on a carbon paper (SigracetGDL29BC) electrode and dried overnight. A small window was cut out on the cathode side and sealed with Kapton film to allow fluorescence signals to pass from the electrode to the detector. To record the operando XANES spectra, the cell was subsequently filled with electrolyte. 0.1 M KHCO₃ aqueous solution, and bubbled with CO₂ for 30 min before test. The XANES spectra were collected under operando

conditions at open-circuit potential (OCP) and different applied potentials.

Fig. R2 Photographs of the operando XAFS instruments. a Customized H-type electrolytic cell. **b** Devices for operando XAFS experiments. **c** Corresponding electrolytic cell.

(2) Results and discussion (corresponding Figures have been added to the Supplementary Information, Supplementary Fig. 23 and 24)

In order to detect the surface structural information during CO₂RR, the operando XAFS spectroscopy of TeN₂-CuN₃ DAC at Cu K-edge was performed using an electrochemical cell. The operando XAFS data were collected at the following four potentials: open-circuit potential (OCP, where the system remains unreacted), and a negative potential (-0.25 vs. RHE), the optimum potential (-0.65 V vs. RHE), and the most negative potential (-1.05 V vs. RHE). As shown in Supplementary Fig. 23, the absorption edge is located at higher energy under OCP, indicating the higher Cu oxidation state. With more negative potentials, the absorption edge shifts to lower energy, implying the decrease of the Cu oxidation state of the Cu. Nevertheless, the absorption edges in XAFS spectra were almost recovered after the applied potential was removed, indicating the structural stability of TeN₂-CuN₃ DAC under CO₂RR potentials (Supplementary Fig. 23 c and d). In spite of our efforts to obtain XAFS data

with higher signal-to-noise ratios, the signal-to-noise ratio is not high enough to reveal the quantitative $\chi(R)$ space and Fourier transform (FT) k^2 -weighted function $\chi(k)$ spectra from extended X-ray absorption fine structure (EXAFS).

In addition, after the stability test, the morphology of TeN₂-CuN₃ DAC was re-examined by HAADF-STEM. Supplementary Fig. 25 shows the bright spots with a high areal density, indicating Te and Cu are still atomically dispersed on the support. The Te-Cu dual-atom pairs can be clearly identified by examining the corresponding intensities of the spots (Supplementary Fig. 25, top right). Moreover, ~80% of the spots were found to be dual-atom sites (Supplementary Fig. 25, bottom right), close to that of original TeN₂-CuN₃ DAC (Fig. 1g), further demonstrating the TeN₂-CuN₃ DAC is stable after CO₂RR.

Similarly, the operando XAFS data for CuN₄ SAC were also collected under the same experimental condition. As shown in Supplementary Fig. 24, the absorption edge at Cu K-edge shifts to lower energy at more negative potentials, and the absorption edges spectra were almost recovered after the applied potential was removed, indicating the structure of CuN₄ SAC is also stable during the CO₂RR process.

The XAFS measurements for Te K-edge require an excitation source of much higher energy (31814 eV), which is currently unavailable, as the synchrotron radiation facility for such high excitation source is currently under maintenance, and the available beamline time has not been determined yet. We hope the reviewer could understand this.

From the above results of operando XAFS spectra of TeN₂-CuN₃ DAC at Cu K-edge under CO₂RR potentials and the structural characterization of TeN₂-CuN₃ by HAADF-STEM image after the catalytic reaction, we can conclude that the structure of TeN₂-CuN₃ DAC is stable under actual CO₂RR potentials.

We have added the above results and discussion in the revised manuscript (Page 13).

Supplementary Fig. 23 Operando XAFS spectra of $\text{TeN}_2\text{-CuN}_3$ DAC at Cu K-edge under CO_2RR in CO_2 -saturated KHCO_3 . a Potential dependence of operando Cu XAFS spectra. **b** Magnified absorption edge of XAFS region. Absorption edges under OCP condition **(c)** and after operando test **(d)**.

Supplementary Fig. 24 Operando XAFS spectra of CuN_4 SAC at Cu K-edge under CO_2RR in CO_2 -saturated KHCO_3 . a Potential dependence of operando Cu XAFS spectra. **b** Magnified absorption edge of XAFS region. Absorption edges under OCP condition **(c)** and after operando test **(d)**.

Supplementary Fig. 25 HAADF-STEM image of $\text{TeN}_2\text{-CuN}_3$ DAC after CO_2RR , with a typical Te-Cu dual-atom spot (top right) and statistical percentages for dual-atom and single-atom sites (bottom right).

We would like to thank all three reviewers for their professional comments and suggestions, which are very helpful to improve the overall quality of this paper. All the changes made in the revised manuscript were highlighted in yellow color.

REVIEWERS' COMMENTS

Reviewer #1 (Remarks to the Author):

The authors have responded to my concerns and made accurate revisions. The manuscript can be accepted.

Reviewer #2 (Remarks to the Author):

The authors have addressed all the concerns I raised and now the current version is recommended to be accepted.

Reviewer #3 (Remarks to the Author):

The authors have added vital experimental data, viz. XAFS measurements under applied electrochemical potentials and post-catalysis HAADF-STEM images, as evidences for the putative existence of the "double-atomic site (DAC) catalyst".

The point-by-point responses and revisions are adequately implemented in the manuscript. The burden of establishing whether the surface states of the catalyst are identical for all characterization methods at a given potential remains heavy and challenging, but the data presented are illustrative of the exquisite surface structure-composition-reactivity control at the atomic level.

The submission is recommended for publication.